# Epidemiological Analysis of Avian Reovirus in China and Research on the Immune Protection of Different Genotype Strains from 2019 to 2020

**DOI:** 10.3390/vaccines11020485

**Published:** 2023-02-20

**Authors:** Dong Liu, Zhong Zou, Shanshan Song, Hongxiang Liu, Xiao Gong, Bin Li, Ping Liu, Qunyi Wang, Fengbo Liu, Dongzu Luan, Xiang Zhang, Yuanzhao Du, Meilin Jin

**Affiliations:** 1State Key Laboratory of Agricultural Microbiology, Huazhong Agricultural University, Wuhan 430070, China; 2Hubei Jiangxia Laboratory, Wuhan 430200, China; 3Key Laboratory of Development of Veterinary Diagnostic Products, Ministry of Agriculture, Wuhan 430070, China; 4YEBIO Bio-Engineering Co., Ltd. of Qingdao, Qingdao 266032, China

**Keywords:** avian reovirus virus, genetic evolution, challenge protection

## Abstract

Avian reovirus (ARV) is the primary pathogen responsible for viral arthritis. In this study, 2340 samples with suspected viral arthritis were collected from 2019 to 2020 in 16 provinces of China to investigate the prevalence of ARV in China and to characterize the molecular genetic evolution of epidemic strains. From 113 samples analyzed by RT-PCR, 46 strains of avian reovirus were successfully isolated and identified. The genetic evolution of the σC gene showed that 46 strains were distributed in 1–5 branches, with the largest number of strains in branches 1 and 2. The σC gene homology among the strains was low, with approximately 62% homology in branches 4 and 5 and about 55% in the remaining branches. The strains circulating during the ARV epidemic in different provinces were distributed in different branches. The SPF chickens were immunized with inactivated vaccines containing strains from branches 1 and 4 to analyze the cross-immune protection elicited by different branches of ARV strains. A challenge protection test was performed using strains in branches 1, 2, 4, and 5. Our results showed that inactivated vaccines containing strains from branches 1 and 4 could fully protect from strains in branches 1, 4, and 5. The results of this study revealed the genetic diversity among the endemic strains of ARV in China from 2019 to 2020. Each genotype strain elicited partial cross-protection, providing a scientific basis for the prevention and control of ARV.

## 1. Introduction

Avian reovirus (ARV) can cause a variety of diseases in poultry, primarily viral arthritis, tenosynovitis, and short stature syndrome, among which viral arthritis and tenosynovitis are the most common [1,2]. ARV belongs to the genus Orthoreovirus. The viral genome consists of segmented double-stranded RNA with ten segments [2]. The virus was first isolated from the respiratory tract of chickens in 1954 and was first identified as a cause of viral arthritis in 1973 [3,4].

ARV can infect a variety of poultry, including chickens, turkeys, ducks, and geese, and it is distributed worldwide [5,6,7]. If the host is infected with ARV, it can damage the immune system, increase susceptibility to other pathogenic microorganisms, raise the mortality rate, decrease the feed conversion rate, and reduce egg production, leading to huge economic losses to the poultry industry [4,8]. ARV was initially found in China in 1958, and its isolation has been reported [2,4]. In recent years, this disease has been regionally prevalent in white-feather broiler farms in some parts of China, showing previously unseen clinical manifestations. Zhang et al. (2019) isolated the ARV variant from broilers with clinically diagnosed arthritis/tenosynovitis in large-scale farms in China from 2013 to 2016. Genetic evolution and pathogenicity analyses show that these strains differ greatly from the vaccine S1133 strain and have strong pathogenicity [9]. In 2017, Chen et al. (2019) isolated a new LY383 strain from immunized broilers. Genomic sequence and phylogenetic analyses based on the sequence of σC nucleotides and amino acids indicated that the isolated strain significantly differed from most Chinese wild strains or commercial vaccine strains currently available on the NCBI database. Challenge protection experiments showed that the existing commercialized vaccines (S1133, 1733, and T98) do not protect against LY383 infection, indicating that the current epidemic strains of ARV in China have been mutating [10]. The rapid accumulation of point mutations, the antigen drift caused by gene recombination, and the widespread use of vaccines have led to the emergence of new genotypes of ARVs. Currently, vaccine failure occurs frequently. However, only a few published ARV sequences are currently available on the NCBI database, and research on whether existing vaccines can protect from new ARV genotypes is limited. Therefore, in this study, we collected the tendons and synovial fluids of chickens suspected to be infected with ARV from 2019 to 2020 for pathogenic testing. The σC genes of some of the isolated ARVs were further sequenced and analyzed. To verify the ARV infection in Chinese provinces from 2019 to 2020, σC gene mutations were analyzed to enrich the ARV bio-information database in China. Each genotype epidemic strain was then isolated and identified. A challenge protection test was performed for the inactivated vaccine with strains from different branches to explore whether the vaccine can protect against the infection of different genotypes of ARV, guiding the prevention and control of ARV.

## 2. Materials and Methods

### 2.1. Source of Samples

Samples of tendons, synovial fluid, and joint swabs from chickens with suspected ARV from 16 provinces and cities in China (Hebei, Henan, Jiangsu, Anhui, Shandong, Hubei, and Jilin) were collected from 2019 to 2020.

### 2.2. Major Reagents and Experimental Animals

RNA extraction kits were purchased from Tianenze Bio, Evo M-MLV one-step RT-PCR kits were purchased from Hunan Ekerui Bioengineering Co., Ltd. (Shenzhen, China), and DNA Marker DL2000 was purchased from Bao Bioengineering (Dalian, China) Co., Ltd. The new Zhifaguan quadruple vaccine (customized vaccine containing viral arthritis S1133 and inactivated vaccine WF17) was obtained from Qingdao Yibang Bioengineering Co. Ltd. (Qingdao, China). Ten-day-old SPF chicken embryos and 50-day-old SPF chickens were purchased from Beijing Veitong Lihua Experimental Animal Technology Co., Ltd. (Beijing, China). The leghorn male hepatocellular (LMH) passage cell line was purchased from the China Veterinary Drug Inspection Institute. The strains used for the challenge test, ARV 1774 and ARV 10415, were isolated and identified in this study; the TS18 and WF17 strains were supplied by the Poultry Research Institute of Shandong Academy of Agricultural Sciences.

### 2.3. Primers

Based on the ARV-related gene sequence published by NCBI, detection and sequencing primers for amplifying the full-length σC gene were designed using Olligo 7 software, with the following detection primers: upstream: 5′-CCCGTCCCTTTCCTCATGCC-3′; downstream: 5′-CAGTAGCGGTGATAGGAGGTGT-3′ (the amplified fragment was 283 bp); upstream: 5′-TGTCAGGCAGCTCAGAACAC-3′; downstream: 5′-CTTAGGTGTCGATGCCGG-3′ (the amplified fragment was 1088 bp). All primers were synthesized by Sangon Biotech Co. Ltd. (Shanghai, China).

### 2.4. Extraction of Viral RNA

Viral RNA was extracted with the Tianenze RNA Extraction Kit. Briefly, 200 μL of tissue grinding fluid was thoroughly mixed with 1000 μL of solution A; 200 μL of chloroform solution was added, and the mix was shaken and centrifuged at 12,000 rpm for 10 min at 4 °C. Then, 600 μL of the aqueous phase was carefully collected and added to a new 1.5 mL centrifuge tube treated with DEPC. An equal amount of solution B was added to the centrifugal tube, which was placed in the centrifugal adsorption column and centrifuged twice at 12,000 rpm for 30 s at 4 °C. Next, the penetrating solution was discarded, 700 µL of the washing solution was added, and the tube was centrifuged at 12,000 rpm for 30 s. After discarding the penetrating solution, the empty tube was centrifuged at 12,000 rpm for 30 s. The spin column was transferred to a new 1.5 mL centrifuge tube treated with DEPC, and 60 µL of the eluent was added and centrifuged at 12,000 rpm for 30 s. Finally, RNA was collected and stored at −20 °C for later use.

### 2.5. RT-PCR Amplification and Sequencing

RT-PCR was conducted using the Evo M-MLV one-step RT-PCR kit under the following reaction conditions: 50 °C for 30 min, 94 °C for 3 min, 94 °C for 30 s, 55 °C for 30 s, 72 °C for 1 min, 35 cycles, and 72 °C for 10 min. The PCR products were separated by electrophoresis on 1.0% agarose gel. The PCR products of positive samples, purified using the BioSpin Gel Extraction Kit, were TA-cloned into the pMD™ 19-T carrier. The connection product was passed to DH5α competence. For the positive bacterial solutions identified by PCR, four clones were selected and sequenced by Shenggong Bioengineering Co., Ltd. (Shanghai, China).

### 2.6. Genetic Evolution Analysis

The sequencing results were compared with the complete σC gene and the chicken-sourced reovirus sequences available in GenBank. Using MEGA6.0 for the genetic evolution analysis, the bootstrap test neighbor-joining method was used to build an evolutionary genetic tree (operations were repeated 1000 times).

### 2.7. Isolation and Identification of ARV

Penicillin-streptomycin (1000 U) was added to the grinding fluid of the tissue samples identified as ARV-positive by RT-PCR. Then, centrifugation was performed at 5000 g/min at 4 °C for 15 min. After sterilization, the supernatant was collected to inoculate LMH cells using a 0.22 μm sterile filter. The cells were cultured at 37 °C and 5% CO_2_ for 5–7 days. Uninfected normal cells were used as controls. The cytopathic conditions were observed and recorded daily. The cells were harvested and frozen when the pathological changes reached 70–80%. When no pathological changes were observed, the cells were allowed to spread continuously for three generations in order to observe possible pathological changes. The cell culture was frozen and thawed three times. RNA was extracted, and RT-PCR was performed using sequencing primers.

### 2.8. Determination of the Embryonic Half Lethal Dose (ELD_50_)

The cell culture seed virus was diluted 10-fold with a stroke-physiological saline solution. Five set dilutions, from 10^−3^ to 10^−7^, were prepared. The virus solution of each dilution was inoculated into six 10-day-old SPF chicken embryos through the chorioallantoic membrane at an inoculation amount of 0.1 mL/piece. The chicken embryos were placed in an incubator at 37 °C. The chicken embryos that died within 24 h were discarded. The embryos were photographed two to three times per day, and observations were performed for five consecutive days. The death of the chicken embryos was recorded daily. The ELD_50_ was calculated using the Reed–Muench method.

### 2.9. Challenge Protection Test

A total of 45 50-day-old SPF chickens were collected and randomly divided into one to nine groups, with five chickens in each group, where one to four were immune groups and five to nine were control groups. According to the vaccine instructions, each chicken in the immune group was injected with 0.2 mL of the vaccine into the neck. The chickens in the control groups were injected with 0.2 mL of sterile normal saline subcutaneously into the necks. After 21 days of vaccination, each challenge strain was injected with 0.1 mL of ELD_50_ (10^2.7^) through the footpads. For Groups 1 and 5, ARV 1774 strains (branch 1) were used for the viral challenge; for Groups 2 and 6, ARV 10,415 strains (branch 2) were used for the viral challenge; for Groups 3 and 7, ARV TS18 strains (branch 5) were used for the viral challenge; for Groups 4 and 8, ARV WF17 strains (branch 4) were used for the viral challenge. Group 9 was the negative control group and was injected with 0.1 mL of sterile physiological saline through the footpads. Clinical examinations of the footpads, toe joints, and tarsal joints were carried out on the 3rd, 5th, and 7th day after the viral challenge, and the results were recorded (see Table 1 for specific methods).

### 2.10. Ethics Statements

All animal experiments were approved by the Research Ethics Committee of Huazhong Agricultural University, Hubei, China (HZAUMO2020-0015). All animal experiments were conducted in accordance with the recommendations in the Guide for the Care and Use of Laboratory Animals from the Research Ethics Committee, Huazhong Agricultural University, Hubei, China.

## 3. Results

### 3.1. Genetic Typing of Chinese ARV Isolates and Homology Analysis of σC Genes from 2019 to 2020

From 2019 to 2020, 113 ARV-positive samples were detected by RT-PCR among 2340 samples, and 46 ARV strains were isolated from LMH cells. A total of 44 of the 46 strains were from different farms, including two strains from different buildings on the same farm (Table 2). Using MEGA 6.0 software, a phylogenetic tree was constructed according to the σC genes of ARV. The 46 strains isolated in this study were genetically divided into six branches based on the ARV classification criteria of Palomino-Tapia et al. [11]. As shown in Figure 1, ARV isolates were distributed in four branches of clusters 1, 2, 3, 4, and 5 from 2019 to 2020. Moreover, ARV strains isolated from different provinces belonged to different branches (see Table 3 for the homology analysis among strains from different branches).

A total of 18 isolates were included in Cluster 1 and distributed in five provinces: Shandong, Liaoning, Fujian, Heilongjiang, and Anhui. The 18 isolated strains had 74–99.8% homology with σC genes, which were divided into two small branches in the evolutionary tree: one (20199330) with 99% homology with S1133 in branch 1, and 17 with a relatively high homology of 84.3–96.3% with the Hungarian strain 4599-V-04 (login number: KX398298) isolated in 2004 [12]. Therefore, this branch was referred to as a 4599-like strain. The homology of the σC gene among strains in this branch with that of the vaccine strains S1133 and WF17 was 75.5–76.2% and 55.3–56%, respectively.

Fourteen isolated strains were included in Cluster 2 and distributed in seven provinces, namely, Shandong, Jilin, Anhui, Fujian, Hebei, Jiangsu, and Shenyang, with 73.3–99.6% homology with σC genes. In the evolutionary tree, strains were divided into two small branches. One branch had a homology of 90.4–91.1% with a U.S. isolated strain (login number: KJ879674) uploaded to the NCBI in 2015, which was referred to as a 98355-like strain. The other branch had a homology of 90.4–91.1% with an isolated U.S. strain (login number: KJ879676) uploaded to the NCBI in 2015, and it is known as a 98856-like strain. The homology of the σC gene of isolated strains in the Cluster 2 branch with that of vaccine strains S1133 and WF17 was 56.5–58.8% and 55.0–56%, respectively.

Cluster 3 comprised four strains distributed in Shandong and Liaoning. However, the homology of the σC genes of the four strains varied widely from 68.7% to 98%. The strains 20,193,634 and 20,209,316 were isolated from different farms in Yantai, Shandong Province, and the σC gene nucleotide homology was 98%. The homology with 99,848 (login number: KJ879690) in NCBI was 91.4–91.6%. The 99,848 strain was isolated from clinically symptomatic tendon synovitis samples in the United States in 2015, while the 20,199,016 and 202,010,268 strains were isolated from different farms in Liaoning Province. The two strains had a 94.5% homology with the 16-0753A/16 (login number MG822677) strain in NCBI (88.6–89.8%). The 16-0753A/16 strain was isolated from viral arthritis samples in Canada in 2018. The homology of the σC gene of the isolated strains in Cluster 3 with that of the vaccine strains S1133 and WF17 was 54.8–56.4% and 55.7–55.9%, respectively.

Three strains were included in Cluster 4, and all were isolated from different farms in Shandong Province. The homology of nucleotides in the σC gene was 97.1–97.4%, with the highest homology being 91.1–91.6% with the Taiwan strain TW-918 (login number: AF297215) in NCBI, 90.8–91.1% with the vaccine strain WF17, and 56–56.3% with the vaccine strain S1133.

Seven strains were included in Cluster 5 and were distributed across three provinces: Shandong, Hebei, and Jiangsu. The homology of the nucleotides in the σC gene was 94.9–100%, with the highest homology being up to 95.6–97.3% with LKYG (login number: MK189466) in NCBI isolated from Shandong, 52.8–53.1% with the vaccine strain S1133, and 61.6–62.7% with the vaccine strain WF17.

### 3.2. Isolation and Identification of ARV Strains

Samples identified as positive by the RT-PCR detection primer (Figure 2) were inoculated into LMH cells. With an increase in the number of passages, the time of pathological changes gradually advanced, and pathological changes were evident. During generations 8–12, the time and characteristics of the pathological changes were stable. After 72 h of inoculation, 70–80% of the cells showed pathological changes, and a large number of cells wrinkled and gathered in groups. After 96 h, the cells began to disintegrate and fall off. The cell culture was frozen and thawed three times, and RT-PCR and sequencing were performed. After the comparison with NCBI BLAST, the cells were found to be ARV-positive.

### 3.3. ELD_50_ of Challenge Strains ARV 1774, ARV 10415, TS18, and WF17

The isolated strains were diluted 10-fold and inoculated into 10-day-old SPF chicken embryos. The mortalities of the diluted chicken embryos are shown in Table 4. Using the Reed–Muench method, the ELD_50_ of each strain was calculated: ARV 1774 strain (10^−5.5^ ELD_50_/0.1 mL), ARV 10,415 strain (10^−5.5^ ELD_50_/0.1 mL), ARV WF17 strain (10^−6.0^ ELD_50_/0.1 mL), and ARV TS18 strain (10^−4.0^ ELD_50_/0.1 mL).

### 3.4. Results of the Challenge Protection Test

In this study, ARV 1774, ARV 10415, WF17, and TS18 of branches 1, 2, 4, and 5 were isolated for challenge protection experiments to evaluate the immune effects of inactivated vaccines containing the ARV S1133 and WF17 strains for branches 1 and 4, respectively. After 7 days of the challenge, the chickens in Groups 5–8 in the drug control group were all affected, and the footpads were notably swollen. However, the chickens in the negative control group were normal. No pathological changes were observed in immune groups 1, 3, and 4, with an immune protection rate of 100%. However, after 3 days of the challenge, four chickens showed pathological changes in Group 2, and the footpads were evidently swollen, with an immune protection rate of only 20%. The results of the challenge protection are shown in Table 5, and the clinical symptoms are shown in Figure 3.

## 4. Discussion

ARV infection has brought great economic losses to poultry farming, emphasizing the importance of continuous research on the prevalence, genetic characteristics, and pathogenicity evolution of emerging ARV strains [13,14]. New ARV strains were discovered in South Korea in 2014 [15]. In addition, several new ARV strains were isolated from broiler flocks vaccinated against ARV in North America in 2016, and several new ARV strains were isolated in western Canada from 2012 to 2017 [11]. However, compared to the abundant information on ARV strains in other countries, few new ARV strains have been identified in China. From 2019 to 2020, 2340 samples of suspected ARV infection were collected from 16 provinces in China, and 113 ARV samples were identified as ARV-positive by RT-PCR; 46 ARV strains were successfully isolated, enriching the Chinese ARV virus library. The hosts infected by these positive samples were primarily white-feathered broilers. ARV strains were not detected in laying hens or yellow-feathered broilers. The time of onset was primarily 15–45 days. The main provinces affected were in eastern China, including Shandong, Jiangsu, Anhui, Fujian, Hebei, Henan, Heilongjiang, Jilin, and Liaoning.

The σC protein is the most variable protein in ARV, located on the surface of the viral capsid. It can produce group-specific neutral antibodies that play an important role in the infection and pathogenicity of ARV [16]. Therefore, according to the σC gene sequence of different ARV strains, the relationship between the difference in sequences and the pathogenicity and immunogenicity of the strains can be analyzed, and the cross-protection of antibodies among different strains can be studied [10]. Sequence homology analysis was performed based on σC. The results showed that, in addition to the 20,199,330 strain, the homology of the other 45 strains with S1133 was relatively low at 52.8–76.2%, and among the strains, a large homology span of 53–100% was observed. Genetic evolution analysis of the σC gene revealed that 46 strains were distributed in more branches. At present, the classification criteria used in various studies are different. For instance, in 2015–2017, American and Canadian scholars divided ARV into clusters I–VI [17,18]; in 2018, Canadian scholars divided ARV into Clusters 1–6 [11]; and scholars at Shandong Agricultural University divided ARV into lineages 1–6 [10]. Combining the different standards in the literature and the ARV strains of this study, according to the typing standards of Canadian scholars in 2018, the 46 isolated strains of this study were distributed in branches 1, 2, 3, 4, and 5. Many isolated strains were found in branches 1 and 2, and the homology of the σC gene with the isolated strains in branches 1, 2, and 3 was high. However, in branches 4 and 5, the homology of the σC gene exceeded 90%. The homology of the strains in branches 4 and 5 was low, accounting for approximately 62%, while the homology in other branches was approximately 55%.

After downloading the sequences of the σC gene of Chinese ARV strains from GenBank, BLAST analysis on the σC genes of the isolated strains was performed using NCBI. The Chinese ARV strains uploaded to GenBank were distributed in branches 1, 4, and 5, whereas, in this study, most of the isolated strains in branch 1 were not homologous with the Chinese strains; however, such isolated strains had a high homology with the 4599-V-04 strain isolated from Hungary. The strains in branches 4 and 5 were homologous with the ARV strains in branches 4 and 5, which were isolated from China and uploaded to GenBank, and the homology was as high as 91.1–97.3%. The strains isolated in this study in branch 2 were homologous to those isolated from the United States. The strains in branch 3 and those isolated from the United States and Canada were observed in branch 1. In 2019–2020, the ARV epidemic strains in China changed considerably, which are different from the wild strains and commercialized vaccine strains in China reported in the NCBI database, with diverse and complex genetic features. 

ARV infections in chickens have been increasingly detected in China since 2008, and an increasing number of ARV field isolates have been isolated from different provinces in China. ARV can cause different clinical symptoms, such as arthritis, respiratory disease, malabsorption syndrome, immunosuppression, etc. It has the characteristics of fast transmission and wide incidence, and chickens of different ages and breeds can be infected, especially for commercial broilers [19,20]. Currently, ARV is mainly controlled through immunization, environmental disinfection, the improvement of feeding management, the strengthening of immunization testing, etc. Vaccines prepared by the traditional vaccine strain S1133 are mostly used in the prevention and control of ARV in the market. However, the incidence of viral arthritis remains high, and the prevalence of ARV is expanding. Consequently, the emergence of more and more new strains has made the disease prevention and control more difficult, and the traditional vaccines are unable to provide better protection against emerging prevalent strains. In this study, two genotypes of the prevalent strains were screened to prepare a multivalent vaccine that could provide good protection against the 1, 4 and 5 branches of the prevalent strains in China, which provided an effective vaccine solution to the problem of multi-gene ARV prevention and control in China’s actual production, thus reducing the economic losses brought by ARV to the farming industry.

In this study, inactivated vaccines containing strains from branches 1 and 4 were used to immunize chickens in order to investigate the cross-immune protection of different genotype ARV strains in China. Then, strains in branches 1, 2, 4, and 5 were used for the challenge. The results showed that all the chickens in the challenge control group were affected, and some were affected in the immune group 2. The footpads of these chickens were slightly swollen, and the internal organs were normal. The chickens in other immune groups were unaffected, indicating that the vaccines in branches 1 and 4 could protect against strains from branches 1, 4, and 5, but not the strains from branch 2, which is also consistent with the homology of the σC gene. The homology in branch 2 was lower than that in branches 1 and 4, whereas that in branches 4 and 5 was higher. However, the incidence sites of SPF chickens were almost limited to the footpad and did not spread to the leg joints. Hyperplasia or purulent secretions were observed only in the affected areas, and no lesions were observed in internal organs. The affected sites gradually faded 7–9 days after the challenge, and slight deswelling was observed in the skin of swollen areas. Furthermore, although the mildly affected area recovered by the 28th day, the affected areas were not restored.

In this study, the prevalence of ARV in China from 2019 to 2020 was investigated, and 46 ARV strains were isolated and identified. Compared with S1133, 45 strains had large variations in the σC gene. They were distributed in different branches of clusters 1, 2, 3, 4, and 5, indicating that the ARV strains in China recently underwent significant genetic changes. The inactivated vaccines with strains from branches 1 and 4, which were selected in this study, offered a good immunotype, with a protection rate of up to 100% for strains in branches 1, 4, and 5.

## 5. Conclusions

Our results revealed that the strains of avian reovirus endemic to China in 2019–2020 were genetically diverse. Each genotype strain elicited partial cross-protection, which provided a scientific basis for preventing and controlling avian reovirus.

## Figures and Tables

**Figure 1 vaccines-11-00485-f001:**
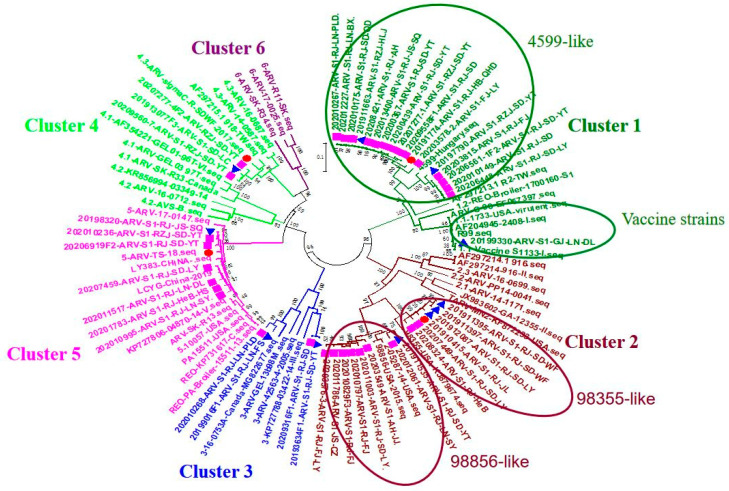
Phylogenetic tree of Avian reovirus virus (ARV) strains based on the σC sequence variability. 
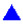
 ARV strains isolated in 2019; 
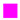
 ARV strains isolated in 2020; 
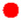
 challenge virus.

**Figure 2 vaccines-11-00485-f002:**
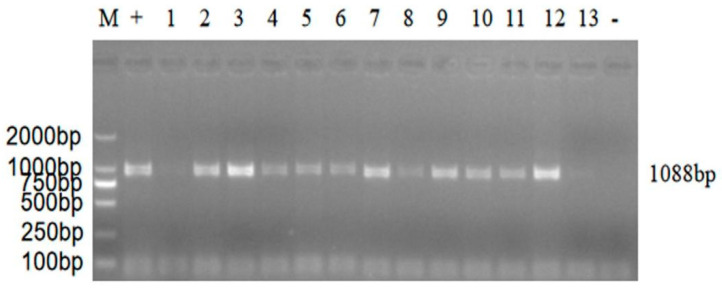
RT-PCR results of Avian reovirus virus (ARV)-infected LMH cells. M: DL2000 Marker; 1–13: ARV isolates; +: Positive control; -: Negative control.

**Figure 3 vaccines-11-00485-f003:**
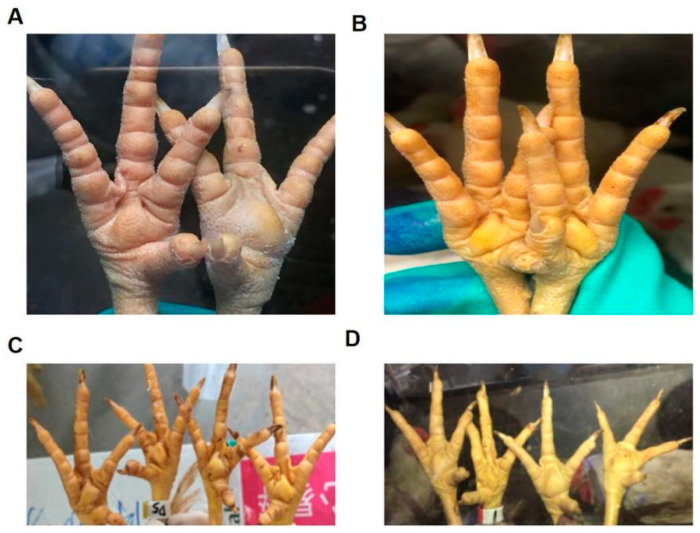
Clinical symptoms in each group after the challenge. (**A**) The footpads of the challenge control group were swollen; (**B**) The footpads of the negative control group were developing normally. (**C**) The footpads of vaccinated group 1 were not evidently swollen; (**D**) The footpads of vaccinated group 2 were obviously swollen.

**Table 1 vaccines-11-00485-t001:** Methods of the challenge test.

Vaccinated Groups		Number	Days	Immunization Dosage	Challenge Virus	Challenge Dosage
Vaccinated groups	1	5	50	0.2 mL	ARV 1774	10^2.7^ ELD_50_
	2	5			ARV 10415	
	3	5			ARV TS18	
	4	5			WF17	
Challenge control groups	5	5	50	0.2 mL	ARV 1774	10^2.7^ ELD_50_
	6	5			ARV 10415	
	7	5			ARV TS18	
	8	5			WF17	
Negative control group	9	5	50	0.2 mL (Normal saline)	Normal saline	0.1 mL

**Table 2 vaccines-11-00485-t002:** The information of 46 Avian reovirus virus isolates. ^※^ Different farms were labeled as A–Z and AB-AT. Farms X1 and X2 have different hen houses than farm X.

Isolates	Origin	Time of Isolation	Breed	σC Genotype	GenBank NO.
20191774	The farm A^※^ in Hebei	2019.3	White-feather broiler	1	ON262159
20197790	The farm B in Shangdong	2019.8	Broiler breeder	1	ON262154
20199330	The farm C in Liaoning	2019.10	White-feather broiler	1	ON262153
20200367	The farm D in Shandong	2020.1	White-feather broiler	1	ON262160
20200439	The farm E in Shandong	2020.1	White-feather broiler	1	ON262161
20203578	The farm F in Fujian	2020.4	White-feather broiler	1	ON262162
201911663	The farm G in Heilongjiang	2019.12	Broiler breeder	1	ON262166
201910415	The farm H in Jilin	2019.11	White-feather broiler	2	ON262181
201911395	The farm I in Shandong	2019.11	White-feather broiler	2	ON262182
201911397	The farm G in Shandong	2019.11	White-feather broiler	2	ON262183
201911535	The farm K in Shandong	2019.11	White-feather broiler	2	ON262173
201912087	The farm L in Shandong	2019.12	White-feather broiler	2	ON262184
20203549	The farm M in Anhui	2020.4	White-feather broiler	2	ON262171
20203576	The farm N in Fujian	2020.4	White-feather broiler	2	ON262172
20193634	The farm O in Shandong	2019.5	White-feather broiler	3	ON262187
20199016	The farm P in Liaoning	2019.9	Broiler breeder	3	ON262185
201913077	The farm Q in Shandong	2019.12	White-feather broiler	4	ON262191
20198320	The farm R in Jiangsu	2019.9	White-feather broiler	5	ON262192
20201783	The farm S in Hebei	2020.3	White-feather broiler	5	ON262193
20203815	The farm T in Fujian	2020.4	White-feather broiler	1	ON262155
20206448	The farm U in Shandong	2020.6	White-feather broiler	1	ON262156
20206561	The farm V in Shandong	2020.6	White-feather broiler	1	ON262157
20206919	The farm W in Shandong	2020.7	White-feather broiler	5	ON262194
20207277-1	The farm X1 in Shandong	2020.7	Broiler breeder	1	ON262163
20207277-4	The farm X2 in Shandong	2020.7	Broiler breeder	4.3	ON262189
20207449	The farm Y in Shandong	2020.7	White-feather broiler	2	ON262179
20207459	The farm Z in Shandong	2020.7	White-feather broiler	5	ON262195
20208324	The farm AB in Hebei	2020.7	White-feather broiler	2	ON262180
20208421	The farm AC in Anhui	2020.7	White-feather broiler	1	ON262164
20209316	The farm AD in Shandong	2020.8	White-feather broiler	3	ON262188
20209558	The farm AE in Shandong	2020.8	Broiler breeder	1	ON262165
20209560	The farm AF in Shandong	2020.8	Broiler breeder	4	ON262190
202010149	The farm AG in Shandong	2020.8	White-feather broiler	1	ON262158
202010175	The farm AH in Shandong	2020.9	White-feather broiler	1	ON262167
202010236	The farm AI in Shandong	2020.9	Broiler breeder	5	ON262196
202010267	The farm AJ in Liaoning	2020.9	White-feather broiler	1	ON262170
202010268	The farm AK in Liaoning	2020.9	White-feather broiler	3	ON262186
202010797	The farm AL in Fujian	2020.9	White-feather broiler	2	ON262174
202010829	The farm AM in Fujian	2020.9	White-feather broiler	2	ON262175
202010995	The farm AN in Liaoning	2020.9	White-feather broiler	5	ON262197
202011003	The farm AO in Shandong	2020.9	White-feather broiler	2	ON262176
202011517	The farm AP in Liaoning	2020.9	White-feather broiler	5	ON262198
202011786	The farm AQ in Jiangsu	2020.9	White-feather broiler	2	ON262177
202012061	The farm AR in Liaoning	2020.9	White-feather broiler	2	ON262178
202012227	The farm AS in Liaoning	2020.1	White-feather broiler	1	ON262168
202013400	The farm AT in Jiangsu	2020.1	White-feather broiler	1	ON262169

**Table 3 vaccines-11-00485-t003:** Homology analysis of the σC gene in different clusters.

Cluster	1	2	3	4	5
1	74–99.8%	58.4–60.2%	54.8–57.2%	53.8–56%	53.1–55.6%
2		73.3–99.6%	51.3–58.3%	54.6–57.8%	53–57.2%
3			68.7–98%	56–56.6%	55.2–58.7%
4				97.1–97.4%	61.6–62%
5					94.9–100%

**Table 4 vaccines-11-00485-t004:** Mortality rate of SPF chicken embryos infected with virulent strains.

ARV1774		ARV10415		WF17		TS18	
Dilution of Virus	Mortality Rate	Dilution of Virus	Mortality Rate	Dilution of Virus	Mortality Rate	Dilution of Virus	Mortality Rate
10^−3^	100%	10^−3^	100%	10^−3^	100%	10^−3^	100%
10^−4^	100%	10^−4^	100%	10^−4^	100%	10^−4^	50%
10^−5^	83%	10^−5^	83%	10^−5^	100%	10^−5^	0
10^−6^	17%	10^−6^	17%	10^−6^	50%	10^−6^	0
10^−7^	0	10^−7^	0	10^−7^	0	10^−7^	0

**Table 5 vaccines-11-00485-t005:** The challenge test results. The vaccinated group challenged with (1) ARV1774, (2) ARV10415, (3) ARV TS18, and (4) WF17. The challenge control group challenged with (5) ARV1774, (6) ARV10415, and (7) ARV TS18. The challenge control group challenged with (8) WF17 and the negative control group.

Vaccinated Groups		Challenge Virus	Challenge Results		
			3-Day	5-Day	7-Day
Vaccinated groups	1	ARV 1774	0/5	0/5	0/5
	2	ARV 10415	4/5	4/5	4/5
	3	ARV TS18	0/5	0/5	0/5
	4	WF17	0/5	0/5	0/5
Challenge control groups	5	ARV 1774	4/5	4/5	5/5
	6	ARV 10415	5/5	5/5	5/5
	7	ARV TS18	3/5	4/5	5/5
	8	WF17	3/5	3/5	5/5
Negative control group	9	\	0/5	0/5	0/5

## Data Availability

Data sharing is not applicable to this article.

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
