# Peer review of "Epidemiological Analysis of Avian Reovirus in China and Research on the Immune Protection of Different Genotype Strains from 2019 to 2020"

_vaccines, 2023, doi:10.3390/vaccines11020485_

Round 1
Reviewer 1 Report
This study describes the epidemiological analysis and Immune protection of different genotype strains of Avian Reovirus in China from 2019-2020. The authors did a lot of work, however, there are still some modifications to be made before accept for publication.
1. Why were only 46 ARV strains isolated from 113 virus-positive samples? I suggest the authors provide a relevant explanation as it is not in the manuscript.
2. The description of the sample collection is not clear. Was each positive sample collected from a different farm, or were multiple samples collected repeatedly from one farm? Please include the background of the sample collection.
3. The description in Figure 3 is ambiguous. Readers cannot intuitively determine which gene nucleotide homology is the virus.
Author Response
1. Why were only 46 ARV strains isolated from 113 virus-positive samples? I suggest the authors provide a relevant explanation as it is not in the manuscript.
A: Thank you for your suggestions. We accepted this comment. Those description have been improved in our new manuscript on line 27-56.
2. The description of the sample collection is not clear. Was each positive sample collected from a different farm, or were multiple samples collected repeatedly from one farm? Please include the background of the sample collection.
A: Thank you for your suggestions. We accepted this comment. Those description have been improved in our new manuscript on line 27-56.
3.The description in Table3 is ambiguous. Readers cannot intuitively determine which gene nucleotide homology is the virus.
A: Thank you for your suggestions. We accepted this comment. The correction have been made in our revised manuscript in Table 3.
Reviewer 2 Report
The manuscript by Dong et al. provides useful updated data on the epidemiology of avian reoviruses in China over 2019-2020 and presents promising data on the immune protection provided by inactivated vaccines.
Interesting data is provided on the circulation of avian reovirus strains across China and how these differ from vaccinate strains including S1133 and the cross-protection provided by vaccines containing certain strains looked to be very promising.
Little information is provided about the inactivated vaccines themselves. I therefore ask the authors to consider this. Looking at the big picture, and given the problem and scale of the disease in China, how feasible and cost-effective would it be to implement a vaccination programme nationwide? Please discuss.
Author Response
The manuscript by Dong et al. provides useful updated data on the epidemiology of avian reoviruses in China over 2019-2020 and presents promising data on the immune protection provided by inactivated vaccines.
Interesting data is provided on the circulation of avian reovirus strains across China and how these differ from vaccinate strains including S1133 and the cross-protection provided by vaccines containing certain strains looked to be very promising.
Little information is provided about the inactivated vaccines themselves. I therefore ask the authors to consider this. Looking at the big picture, and given the problem and scale of the disease in China, how feasible and cost-effective would it be to implement a vaccination programme nationwide? Please discuss.
A: Thank you for your suggestions. We accepted this comment. Those discussion have been added in our revised manuscript on line 336-353.